# MALDI-TOF MS Approaches for the Identification of the Susceptibility of Extended-Spectrum β-Lactamases in *Escherichia coli*

**DOI:** 10.3390/microorganisms11051250

**Published:** 2023-05-09

**Authors:** Yuriko Matsumura, Kazuko Ikegaya

**Affiliations:** 1Postgraduate School of Healthcare, Tokyo Healthcare University, 4-1-17 Higashi-Gotanda, Shinagawa-ku, Tokyo 141-8648, Japan; 2Shizuoka City Shimizu Hospital, 1231, Miyakami, Shimizu-ku, Shizuoka 424-8638, Japan; smz-hp-kensa@bz03.plala.or.jp

**Keywords:** antibiotic resistance, MALDI-TOF MS, extended-spectrum β-lactamase, cefotaxime

## Abstract

The increase in multidrug-resistant microorganisms that produce extended-spectrum β-lactamases (ESBLs) and carbapenemases is a serious problem worldwide. Recently, matrix-assisted laser desorption ionization–time-of-flight mass spectrometry (MALDI-TOF MS) has been used for the rapid detection of antibiotic-resistant bacteria. The objective of this study was to establish a method to detect ESBL-producing *Escherichia coli* by monitoring the hydrolyzation of cefotaxime (CTX) using MALDI-TOF MS. According to the ratio of the peak intensity of CTX and hydrolyzed-CTX-related compounds, the ESBL-producing strains could be clearly distinguished after 15 min of incubation. Moreover, the minimum inhibitory concentration (MIC) values for *E. coli* were 8 μg/mL and lower than 4 μg/mL, which could be distinguished after 30 min and 60 min of incubation, respectively. The enzymatic activity was determined using the difference in the signal intensity of the hydrolyzed CTX at 370 Da for the ESBL-producing strains incubated with or without clavulanate. The ESBL-producing strains with low enzymatic activity or *bla*_CTX-M_ genes could be detected by monitoring the hydrolyzed CTX. These results show that this method can rapidly detect high-sensitivity ESBL-producing *E. coli*.

## 1. Introduction

The increase in antibiotic- and multidrug-resistant microorganisms is a crucial problem worldwide. Bacterial strains possess six kinds of resistant mechanisms: (i) antibiotic inactivation by hydrolysis, (ii) by redox process, and (iii) by group transfer, and (iv) antibiotic resistance via target modification, (v) via genetic mutation, and (vi) via horizontal gene transfer [1,2]. Especially for the Gram-negative bacteria such as *Enterobacterales*, *Pseudomonas* species, and *Acinetobacter* species, the microorganisms producing the extended-spectrum β-lactamases (ESBLs) or carbapenemases cause these crucial problems. Recently, the variety and drug resistance of resistant bacterial species have been increasing. Carbapenem-resistant *Enterobacterales* (CRE) not only produce carbapenemases that confer resistance to carbapenems, but often also acquire resistance to other antibiotics [3]. The order of *Enterobacterales* has plasmidic resistance determinants that are easily transferred among the species. The number of ESBL-producing strains has also been increasing worldwide [4]. The proportion of ESBL-producing *Escherichia coli* is now >50% in Asian countries, such as in China and India. They were widely isolated in Japan in the 2000s [5,6]. As the growing number of antibiotic-resistant microorganisms is the cause of the increase in healthcare problems, the rapid detection and determination of the resistance type is required. In order to prevent the spread of antibiotic- and multidrug-resistant microorganisms, broad-spectrum antimicrobials, which are sometimes administered when causative organisms have not been identified, should be substituted for the appropriate narrow-spectrum antimicrobials as soon as possible. Additionally, empirical antimicrobial therapy should be changed to definitive therapy.

Antibiotic resistance can be detected using classical methods such as the disk diffusion method or microliquid dilution method to determine the minimum inhibitory concentration (MIC), as well as using DNA-based methods performed via real-time PCR in hospital. Antimicrobial susceptibility testing should be carried out within 1 to 3 days as the MIC is determined by monitoring the growth of microorganisms. Moreover, the PCR method can only detect the genes encoding known enzymes. Based on these reasons, new methods for rapid antibiotic resistance detection have been studied and developed. Matrix-assisted laser desorption ionization–time-of-flight mass spectrometry (MALDI-TOF MS) is one of the powerful tools used for the identification of microorganisms. It has been widely used as a routine method all over the world because of its economical and diagnostic benefits. The use of the MALDI-TOF MS assay provides rapid identification results, which can be obtained approximately 24 h faster than those of classical methods. Recently, it was used for the detection of antibiotic resistance by monitoring the hydrolysis of antibiotics [7,8,9,10,11,12,13]. Most research objects that require the use of a MALDI-TOF MS assay are carbapenem-resistant microorganisms. The number of reports focusing on ESBL-producing *Enterobacterales* has increased. β-Lactamase produced from microorganisms hydrolyzes the β-lactam ring of the antibiotic, resulting in an increase in the molecular mass by 18 Da on the mass spectrum. The direct monitoring of the structural change in antimicrobials by MALDI-TOF MS is useful for detecting antimicrobial resistance. Thus, in past studies, we established a rapid detection method for ESBL-producing bacteria using MALDI-TOF MS [14,15]. We confirmed the hydrolysis of cefotaxime (CTX) and the assumed reaction mechanism after hydrolysis of cefotaxime by monitoring the mass spectrometry/mass spectrometry (MS/MS) of each hydrolyzed cefotaxime as precursor ions using MALDI-TOF MS and liquid chromatography–mass spectrometry (LC-MS-IT-TOF) [14]. Moreover, by using clinical isolates of *Escherichia coli*, *Klebsiella pneumoniae*, and *Proteus mirabilis* as tested strains, we developed a rapid detection method for distinguishing the antimicrobial-resistant microorganisms of four kinds of antimicrobials, including cefotaxime (CTX), cefpodoxime (CPDX), piperacillin (PIPC), and cefpirome (CPR) [15]. We describe how the ratio of signals derived from the hydrolysis products divided by the total signal intensity detected is useful for identifying ESBL-producing strains much more rapidly than when using conventional methods. The original mass peak of the antibiotics disappeared, whereas the hydrolyzed peak appeared on the mass spectrum. In order to detect the ESBL-producing microorganisms using MALDI-TOF MS, both the original and hydrolyzed antibiotics were monitored.

Here, we describe the rapid detection of ESBL-producing *Enterobacterales* by focusing on clinically isolated *E. coli* using MALDI-TOF MS to show MIC values from <1 μg/mL to ≥64 μg/mL, as well as the antibiotic plates used daily. CTX was chosen as the antibiotic as it has been widely used in Japan. The relationship between the enzymatic activity of the tested strains and the results of MS analysis in terms of the signal intensities derived from hydrolyzed and unhydrolyzed CTX is also discussed.

## 2. Materials and Methods

### 2.1. Bacterial Strains and Cultivation

*E. coli* NCTC 13462 and ATCC 25922 were used as β-lactamase positive and negative strains, respectively. These strains were subcultured on heart infusion agar (Eikenkagaku Co., Ltd., Tokyo, Japan). After incubation for 16–18 h at 35 °C under the aerobic culture, colonies were picked and suspended in saline at the appropriate turbidity of McFarland standard No. 0.5 for the analysis. Clinical isolates of *E. coli* (Shizuoka City Shimizu Hospital, Shizuoka, Japan), which were identified by MALDI-TOF MS and stored in the skim milk at −80 °C, were subcultured on heart infusion agar. After incubation for 16–18 h at 35 °C, they were used in the same manner for the control strains. The clinical isolates were classified as ESBL-producing strains using the disk diffusion method according to the Clinical and Laboratory Standards Institute (CLSI) guideline M100-S26. The presence of resistance genes was confirmed using PCR. For the disk diffusion method, the turbidity of suspension was adjusted to the McFarland standard No. 0.5 using saline. Then, 30 μg of CTX and 30 μg/10 μg disks of CTX/CLA were placed onto the inoculated Mueller–Hinton agar plate. After incubation for 16–18 h at 35 °C under the aerobic culture, the increase in zone size of more than 5 mm was considered to be positive for ESBL production. For PCR-based detection, the turbidity of suspension was adjusted to the McFarland standard No. 3 using water. For plasmid DNA extraction, the suspension was heated at 100 °C for 15 min and centrifuged at 16,200× *g* for 2 min. The supernatant was subjected to a PCR assay using specific primers as shown in Table 1. Each 20 μL reaction tube contained 1 μL of DNA extract solution, 10 μL of FastStart Essential DNA Green Master, 1 μL of both the 10 μM forward and reverse primers, and 7 μL of DNase- and Rnase-free PCR-grade water, and subjected to the real-time PCR (LightCycler^®^ Nano system, Roche Diagnostics, Mannheim, Germany). PCR was carried out in a thermal cycler and the cycling parameters were as follows: (a) hold at 95 °C for 600 s; (b) 3-step amplification at 95 °C for 10 s, 57 °C for 10 s, and 72 °C for 15 s for 45 cycles; (c) pre-melt hold at 95 °C for 60 s, and (d) melt at 60 °C for 60 s and 97 °C for 1 s. The melting temperature (T_m_) was obtained from analysis of the melting curve using the LightCycler^®^ Nano software Version 1.1.0. The T_m_ of the strains with *bla* _CTX-M_ genes was 85–86 °C for the ESBL-positive strains, as reported by Nass T. et al. [16].

### 2.2. Hydrolysis Assay

In total, 100 μL and 25 μL of *E. coli* suspensions at the appropriate turbidity were placed on the CTX (32 μg) and CTX/CLA (4 μg/4 μg) well of the DPD-1 plate (Eikenkagaku Co., Ltd., Tokyo, Japan). The information about the panel is available in Appendix A. After incubation at 35 °C under the aerobic culture for 15, 30, and 60 min, the supernatants were used in the MALDI-TOF MS analysis. 

### 2.3. MALDI-TOF MS Analysis

MALDI-TOF MS measurements were performed with a VITEK^®^ MS Plus (SARAMIS Premium Database, version: 4.10, system version: 4.0.0.14; Sysmex bioMérieux, Tokyo, Japan). Four μL of the supernatant from the hydrolysis assay was directly spotted onto a steel MALDI target plate. Dried spots were overlaid with 1 μL of 2-cyano-4-hydroxycinnamate (CHCA) solution as a MALDI matrix and air-dried to obtain a measurement sample for MALDI-TOF MS. After drying the matrix, spectra were recorded in the positive linear mode using the following parameters: a mass range from 150 to 1000 Da; an acceleration voltage of 20 kV; 5 shots accumulated per profile; and 100 profiles per sample. All acquired spectra were automatically processed and the ASCII-formatted peak lists were exported. The CTX-related peaks were picked up and their peak intensities were listed using Excel. The CHCA peaks, [M+H]^+^ at 190.05 Da and [2M+H]^+^ at 379.09 Da, were employed for calibration.

## 3. Results

### 3.1. Strain Characterization

The clinically isolated microorganisms are listed in Table 2. The total number of microorganisms clinically isolated from March 2016 to August 2016 was 2620, and the number of *E. coli* strains was 414. Some of the clinically isolated *E. coli* strains (96 that were randomly chosen) were characterized by the disc diffusion method and the PCR-based detection method. Forty percent of the strains were CTX-resistant. Only one strain had the *bla*_CTX-M_ gene which showed CTX sensitivity and was obtained from the results of the disk diffusion method (described as MIC < 1 μg/mL in Table 2). The MIC values for the sensitive strains showed susceptibility to ampicillin (AMP), cefmetazole (CMZ), ceftazidime (CAZ), cefepime (FEP), imipenem (IPM), meropenem (MEM), gentamicin (GEN), amikacin (AMK), levofloxacin (LVX), and ciprofloxacin (CIP). All the CTX-resistant strains had the *bla*_CTX-M_ genes.

### 3.2. Detection of CTX and Hydrolyzed CTX by VITEK^®^ MS plus Using Standard Strains

The molecular weight and exact mass of CTX were 455.47 and 455.06 [17]. The molecular weight and exact mass of the sodium adduct of CTX were 477.45 and 477.04, respectively [18]. The MALDI-TOF MS spectra for CTX incubated with or without strains for 15 min are shown in Figure 1. The molecular peak of CTX ([M+H]^+^) was detectable at ~456 Da with the sodium adducts ([M+Na]^+^) at ~478 Da, as reported previously using VITEK^®^ MS Plus [14] and MALDI Biotyper^®^ [15], when incubated for 60 min. The molecular signal of elimination of the acetyl group of CTX was also detectable at ~396 Da as previously reported [19,20,21,22,23]. The MS spectrum for the supernatant of *E. coli* ATCC 25922, as the β-lactamase-negative strain incubated with CTX, was similar to that for CTX in saline (Figure 1a,b). On the other hand, the incubation of CTX with *E. coli* NCTC 13462, as the β-lactamase-positive strain, results in the appearance of a hydrolyzed CTX-related peak at ~370 Da and ~414 Da (Figure 1c). The ratios of total hydrolyzed CTX signal intensities at ~370 Da and ~414 Da divided by the sum of both of the hydrolyzed CTX and unhydrolyzed CTX signal intensities at ~396 Da, ~456 Da, and ~478 Da were calculated using Equation (1):ratio = sum(hydrolyzed CTX signal intensities)/sum(unhydrolyzed CTX signal intensities and hydrolyzed CTX signal intensities) = (*I*_370_ + *I*_414_)/(*I*_370_ + *I*_414_ + *I*_396_ + *I*_456_ + *I*_478_)(1)
where *I* and the subscript numbers mean the signal intensity and observed mass numbers, respectively. The ratios for the β-lactamase-positive strain (*E. coli* NCTC 13462) and -negative strain (*E. coli* ATCC 25922) were 0.45 and 0.09, respectively, when the *E. coli* suspensions at a concentration of McFarland standard No. 1 were incubated with CTX for 15 min.

The results of using the different concentrations of *E. coli* suspensions with varying incubation times are shown in Figure 2. The ratio for the β-lactamase-negative strain was below 0.11 for all suspension concentrations regardless of the incubation time. The ratio for the β-lactamase-positive strain with the 10-fold diluted suspension of McFarland standard No. 0.5 was above 0.18 after 60 min of incubation, whereas it decreased to be under 0.05 after incubation for more than 120 min. When McFarland standard No. 0.5 or No. 1 suspensions were used for the hydrolysis assay, the ratios after incubation for less than 60 min were below 0.12 for the β-lactamase-negative strain and above 0.35 for the β-lactamase-positive strain. Those for the β-lactamase-positive strain after incubation for more than 120 min decreased.

### 3.3. Detection of Clinically Isolated Strains

The results from the 96 kinds of clinically isolated strains are shown in Figure 3. The ratios calculated using Equation (1) for β-lactamase-positive strains after 15 min of incubation were 0.16 ± 0.08 and were higher than those for β-lactamase-negative strains (0.09 ± 0.02). This phenomenon was also observed after 30 min and 60 min of incubation, where the ratios for β-lactamase-positive strains were 0.17 ± 0.07 for 30 min of incubation and 0.18 ± 0.09 for 60 min of incubation, and those for β-lactamase-negative strains were 0.09 ± 0.02 for 30 min of incubation and 0.07 ± 0.03 for 60 min of incubation.

Focusing on the MIC values lower than 8 μg/mL, the ratios after 60 min of incubation for all strains were above 0.12, as shown in Figure 4. On the other hand, one strain with an MIC value of 4 μg/mL and two strains with MIC values of 8 μg/mL had ratios below 0.10 both after 15 min and 30 min of incubation. 

As the normalized MS intensity was useful for considering the enzyme activity, the peak at 370 Da assigned for the hydrolyzed CTX was focused on. When CLA coexisted with CTX, the β-lactamase activity was inhibited, resulting in the decrease in the signal intensity at 370 Da. The difference in the normalized intensity incubated with or without CLA (*I*_370_(CTX)-*I*_370_(CTX/CLA)) of the ESBL-producing strains showing MIC values ≥64 μg/mL is shown in Figure 5. The average and median decreases in the peak heights were 0.65 and 0.51, respectively. The decrease in six kinds of strains was higher than 0.80, whereas for one strain (number 1) it was 0.29. The clinically isolated strains were categorized into three parts, including the low group: the values were below the median; middle group: the values were between 0.51 and 1.00; and high group: the values were above 1.00. The difference in the peak height for the β-lactamase-producing strain (*E. coli* NCTC 13462) was 1.61, and this strain was categorized in the high group. 

## 4. Discussion

MALDI-TOF MS is a powerful tool for identifying microorganisms because of its speed, reliability, and low cost. Recently, MALDI-TOF MS was utilized for the rapid detection of antibiotic-resistant bacteria [7]. The number of reports on the detection of carbapenemase-producing microorganisms [24] is greater than the number on ESBL-producing microorganisms. Moreover, the number of reports on the detection of *E. coli* is the highest out of all the reports concerning *Enterobacterales*-producing ESBLs, and CTX has been widely used as the therapeutic agent. In this study, we focused on the direct detection of β-lactamase activity based on CTX hydrolysis using MALDI-TOF MS. The signals for CTX were reported to be observed at 396.5 Da, 456.5 Da, and 478.5 Da on the MS spectra, and those for hydrolyzed CTX were observed at 414.5 Da and 370.5 Da using the MALDI Biotyper^®^ with several groups [8,9,10,11,12,13]. When the saline was used for the bacterial suspended solution, a sodium adduct was detected on the mass spectrum. As the exact masses for CTX and the sodium adduct of CTX were 455.06 [17] and 477.04 [18], the signals at 456.5 Da and 478.5 Da were assigned to be [M+H]^+^ and [M+Na]^+^, respectively. A signal observed at 369.5 Da was assigned to be the elimination of the acetyl group of CTX [19]. When the suspension of *E. coli* ATCC 25922 incubated with CTX was used for the analysis, the signals at ~369 Da, ~456 Da, and ~478 Da were observed on the MS spectrum, suggesting that the hydrolysis of CTX by β-lactamase did not proceed. When the suspension of *E. coli* NCTC 13462 with McFarland standard No. 0.5 was incubated for 15 min with CTX, that is, when both of the CTX-related compounds and the hydrolyzed CTX-related compounds existed in the supernatant, the molecular peak of elimination of the acetyl group of CTX ([M-acetyl group+H]^+^) at ~369 Da, CTX ([M+H]^+^) at ~456 Da, the sodium adduct of CTX ([M+Na]^+^) at ~478 Da, and two kinds of hydrolyzed CTX-related compounds can be observed, as previously reported and shown in Figure 1c [14,15]. We previously reported the assumed reaction mechanism after hydrolysis of CTX by MS/MS analysis using MALDI-TOF MS [14]. Enzymatic cleavage of the β-lactam ring is characterized by the addition of a water residue, resulting in a mass sift from 456 Da to 474 Da, which is an increase of 18 Da. The subsequent decarboxylation, deacetylation, and cyclocondensation reactions occurred and the mass signals at ~370 Da were observed. At the same time, hydrolyzed CTX underwent deacetylation and then cyclocondensation, and the mass signals at ~414 Da were observed. Signals at ~370 Da and ~414 Da were observed, suggesting that these assumed reactions occurred. The ratio calculated by using Equation (1) for *E. coli* NCTC 13462 was 0.45 and that for *E. coli* ATCC 25922 was 0.09. These results show the same trend as our previous report that used the MALDI Biotyper^®^ (Bruker Daltonik GmbH, Bremen, Germany), where the signals at ~370 Da, ~456 Da, and 478 Da were used for the calculation [15]. This indicates that our methodology using the signals derived from both hydrolyzed and unhydrolyzed CTX is an instrument-independent method. The increase in enzyme concentration in the system induced the acceleration of the hydrolysis of CTX, which resulted in shortening the incubation time. The turbidity of the *E. coli* suspensions was changed in order to alter the enzyme concentration used in the hydrolysis assay. The threshold time of the detection limit was 120 min for the 10-fold dilution of McFarland standard No. 0.5, and the ratio dramatically decreased for McFarland standards No. 0.5 and No. 1 after 120 min of incubation. These findings indicate that the hydrolyzed CTX was more decomposed, resulting in the decrease in the signal intensity at 370 Da. Thus, we suggest that the incubation time for the hydrolysis assay be set to no longer than 60 min. With 96 strains of clinical isolates, the signal intensities at 370 Da for the β-lactamase-positive strains (39 strains) were higher than those for the β-lactamase-negative strains (57 strains), as shown in Figure 3. Moreover, the ratios calculated using Equation (1) for the β-lactamase-positive strains after 30 min and 60 min of incubation were significantly different from those for the β-lactamase-negative strains. These results indicate that the ratios calculated using Equation (1) were useful for distinguishing ESBL-producing strains, and the threshold ratio for determining whether the tested bacterium is an ESBL-producing strain or not was 0.11. It was previously reported that the incubation time for the rapid detection of Enterobacteriaceae-producing ESBLs by monitoring the structural change in antibiotics using MALDI-TOF MS should be 1–3 h [8,9,10,11,12,13]. Recently, rapid detection was carried out by directly analyzing positive blood cultures. In this case, the incubation times were set for 1 h [21,23], 1.5 h [20,22], and 3 h [19]. In this study, ESBL-producing *E. coli* was distinguishable within a culture time of 15 min using Equation (1), where the ratio was calculated by dividing the sum of the hydrolyzed signal intensities by the sum of the signal intensities of those targeted for observation. Thus, we suggest that a 15 min incubation was enough for distinguishing the ESBL-producing strains. It is possible to apply this method for the 15 min rapid detection of cephalosporin-resistant strains from blood cultures via MALDI-TOF MS. 

The MIC value was affected by the enzymatic activity of β-lactamase or the enzyme concentration in the system. The signal intensity was useful for considering enzymatic activity. As the normalized signal intensities for CTX at 456 Da and 478 Da were reported to show the linear correlations with the CTX content when the matrix signal was used for the internal standard signal [9], the hydrolysis activity determined via the MALDI-TOF MS analysis using the signal intensities for hydrolyzed CTX and nonhydrolyzed CTX was compared to that determined by the MIC assay. When confirming with the CLSI, ESBL production is suspected in strains with an MIC value greater than 2 μg/mL, and thus a phenotypic confirmation test is required. Additionally, strains showing an MIC value equal to 2 μg/mL are judged to be intermediate. The time to provide a determination of antibiotic resistance should be valid in these cases. When the enzymatic activity of β-lactamase is low, the hydrolysis of CTX slowly proceeds, resulting in the increase in the hydrolyzed CTX concentration requiring a longer incubation time. This means that the signal intensity of the hydrolyzed CTX is low and that it takes a longer incubation time to distinguish the ESBL-producing strain. When the ratio calculated using Equation (1) was used for the analysis, it showed that 60 min of incubation was enough to distinguish ESBL-producing strains showing an MIC value above 2 μg/mL. These findings suggest that our methodological approach has great potential for the rapid and highly sensitive detection of ESBL-producing strains against CTX.

When the enzymatic activity of β-lactamase is high, CTX was easily and quickly hydrolyzed, resulting in the increase in the hydrolyzed CTX concentration. As CLA is known to be an inhibitor of β-lactamase, the signal at 370 Da assigned for the hydrolyzed CTX decreased when CLA coexisted with CTX in the hydrolysis assay. The signals at 370 Da were focused on, and enzymatic activity was determined using the difference in the signal intensity for the clinically isolated ESBL-producing strains incubated with or without CLA, determined using Equation (2):I_370_(CTX) − I_370_(CTX/CLA)(2)
where I_370_(CTX) is the signal intensity at 370 Da incubated with CTX and I_370_(CTX/CLA) is the signal intensity at 370 Da with CTX in the presence of CLA. This decrease in the signal intensity indicates the enzymatic activity of the tested strain, as CLA was the inhibitor of β-lactamase activity. When *E. coli* NCTC13462 was used, the decrement value was 1.61, as shown in Figure 5. Two clinical isolates (number 26 and 27 in Figure 5) also showed a decrement value above 1.6. This means that these three strains showed the highest enzymatic activity out of all the tested strains. The decrement values for the clinical isolates showing MIC values above 64 μg/mL, excluding tow strains (a total of 25 strains), were 0.56 ± 0.19, as listed in Table 3.

While the number of strains were limited, it was suggested that the enzymatic activity for the tested strains with MIC values above 64 μg/mL showed a decrease around 0.56. It was suggested that the decrease calculated using Equation (2) reflected the enzymatic activity of the ESBL-producing strain. The decreases for the other strains showing low MIC values are also listed in Table 3. The calculated decrement value tends to decrease as the MIC value for the tested strain becomes low, even though the sample size was limited. Moreover, in the case of the sensitive strain with *bla*_CTX-M_ genes, only one strain was analyzed, and the calculated decrement value was 0.37. This value was larger than that for the strain showing the lowest value of all tested strains with an MIC value ≥64 μg/mL with a calculated decrement value of 0.29. This strain was judged via our methodology using MALDI-TOF MS in order to be resistant, as the MIC value was less than 1. These findings suggest that it is possible to detect the CTX-resistant strains showing *bla*_CTX-M_ gene expression even though the expression levels of genes or enzyme activity are low. Our methodological approach demonstrates a great advantage for detecting strains with the *bla*_CTX-M_ gene or those that show low enzymatic activity. It could be possible to achieve early de-escalation to an appropriate antimicrobial agent if causative bacteria show low-level gene expression and enzymatic activity. As our methodology using MALDI-TOF MS monitored the structural change caused by the enzymatic reaction, the main limitation of our study is that not all of the antimicrobial-resistant bacteria could be detected. As the sample size was limited, further experiments with a larger sample size need to be carried out, and the types of bacteria such as *Proteus mirabilis* and *Klebsiella pneumoniae*, as well as the other types of enzyme genes such as the *bla*_OXA-48_ gene and the number of antibiotics, should be varied in order to use this methodology in hospitals. Moreover, this approach could be extended to other antibiotic agents and microorganisms. 

## 5. Conclusions

Our study shows that our methodology using a classical assay kit and MALDI-TOF MS was effective for the rapid detection of ESBL-producing bacteria. This method could detect resistance against ESBL-producing *E. coli* with a short incubation time of only 15 min and had high sensitivity. The sensitivity of this method was higher than that of the classical disk diffusion method. Our method combined with the identification of microorganisms using MALDI-TOF MS is suggested to be a powerful tool to determine early therapeutic guidance for patients.

## Figures and Tables

**Figure 1 microorganisms-11-01250-f001:**
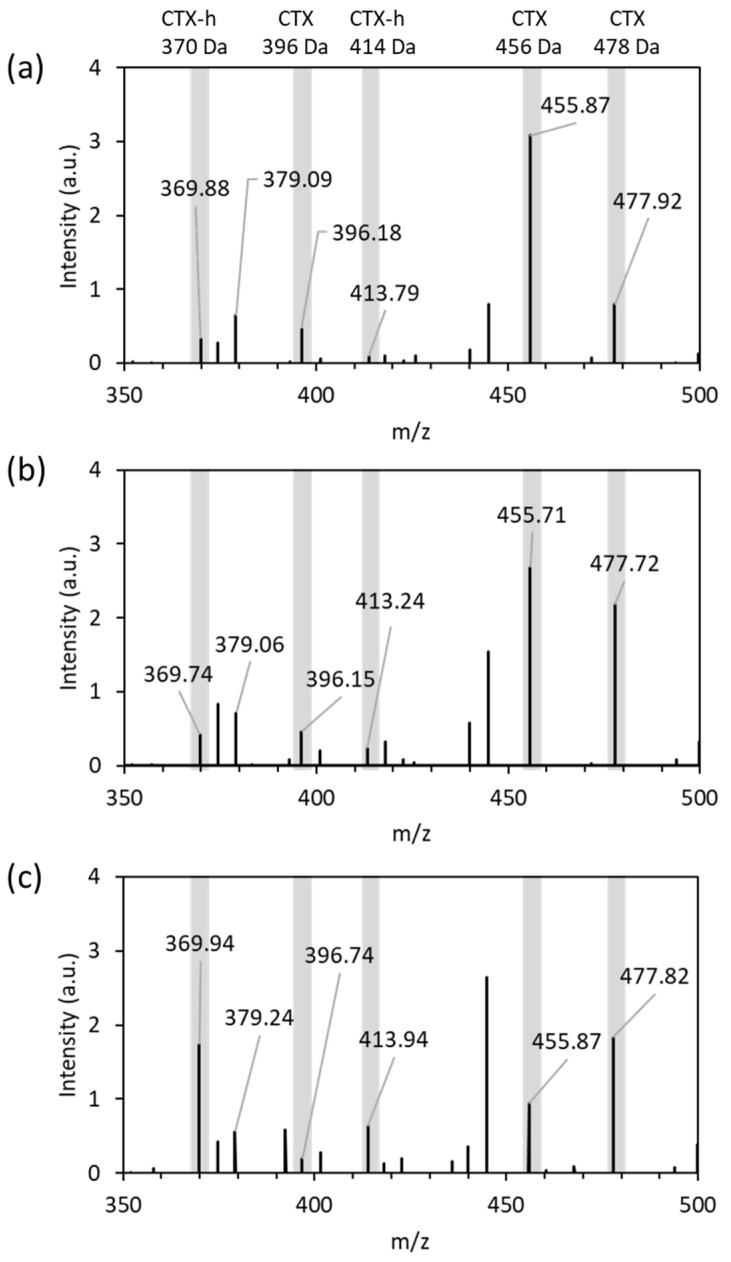
MALDI TOF-MS spectra of cefotaxime after incubation for 15 min: (**a**) in saline without strain; (**b**) with the supernatant of the β-lactamase-negative *E. coli* ATCC 25922 suspension; (**c**) with the supernatant of the β-lactamase-positive *E. coli* NCTC 13462 suspension. Gray bars indicate the unhydrolyzed- signals of CTX (CTX) and hydrolyzed-derived signals of CTX (CTX-h).

**Figure 2 microorganisms-11-01250-f002:**
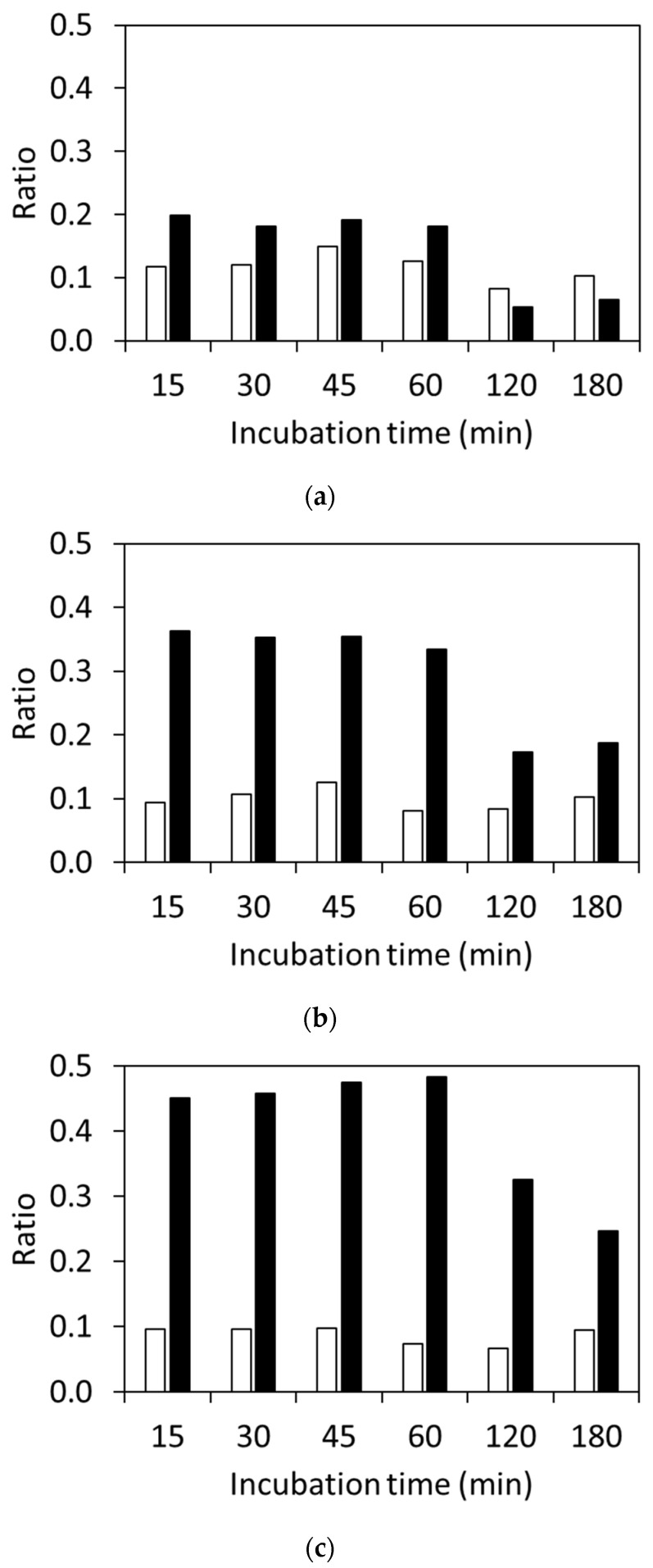
Ratio of sum of hydrolyzed CTX signal intensities at 370 Da and 414 Da divided by the sum of the hydrolyzed CTX and unhydrolyzed CTX signal intensities using Equation (1). The suspension concentrations were (**a**) 10-fold dilution of McFarland standard No. 0.5, (**b**) McFarland standard No. 0.5, and (**c**) McFarland standard No. 1. White bar: *E. coli* ATCC 25922; black bar: NCTC 13462.

**Figure 3 microorganisms-11-01250-f003:**
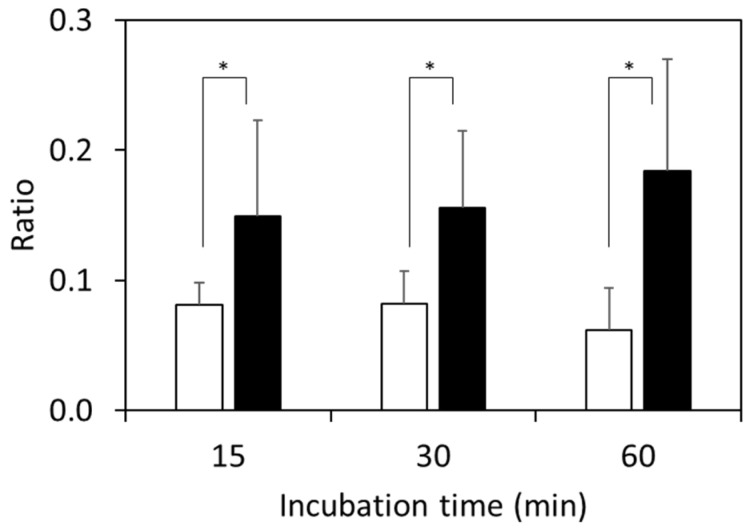
Ratio of sum of hydrolyzed CTX signal intensities at 370 Da and 414 Da divided by the sum of the hydrolyzed CTX and unhydrolyzed CTX signal intensities using Equation (1) for the clinically isolated strains. The suspension concentration was McFarland standard No. 1. White bar: β-lactamase-negative strains incubated with CTX; black bar: β-lactamase-positive strains incubated with CTX. * *p* < 0.01.

**Figure 4 microorganisms-11-01250-f004:**
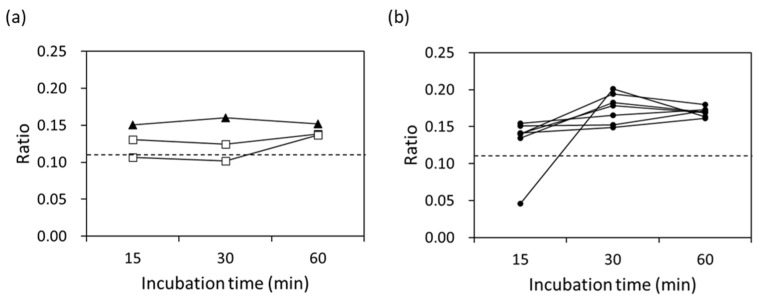
Ratio of sum of hydrolyzed CTX signal intensities at 370 Da and 414 Da divided by the sum of the hydrolyzed CTX and unhydrolyzed CTX signal intensities using Equation (1) for the clinically isolated strains showing MIC values from 2 to 8 μg/mL: (**a**) 2 μg/mL (▲) and 4 μg/mL (□); (**b**) 8 μg/mL (●). The dashed line means the threshold of detection for the ESBL-producing strains.

**Figure 5 microorganisms-11-01250-f005:**
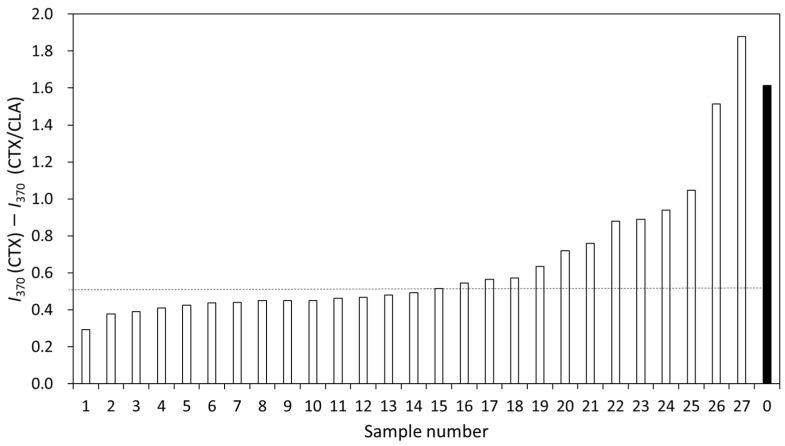
The difference in the normalized peak intensity (bars) at 370 Da for the clinically isolated ESBL-positive strains showing MIC ≥64 μg/mL (white bar) incubated with CTX or CTX/CLA for 15 min. Number 0 means *E. coli* NCTC13462 (black bar). The dashed line means the median (0.51).

**Table 1 microorganisms-11-01250-t001:** Amplification and pyrosequencing primers used in this study [16].

Primer Name	Sequence (5′ to 3′)
CTX-MA1 bio (forward, biotinylated)	biotin-[C/G]C[G/A/C]ATGTGCAG[C/T]ACCAGTAA
CTX-MA5 (reverse)	TG[A/G]G[A/C]AATCA[A/G][C/T]TT[A/G]TTCAT

**Table 2 microorganisms-11-01250-t002:** Characterization of clinically isolated *E. coli* from March 2016 to August 2016.

Bacterial Isolates	Number of Strains
Isolated *E. coli*	414
The strain used for the test	96
*bla*_CTX-M_ detection using real-time PCR assay	40
MIC value for CTX	
<1 μg/mL	1
2 μg/mL	1
4 μg/mL	2
8 μg/mL	7
32 μg/mL	2
≥64 μg/mL	27
Susceptibility to specified antibiotics ^1^	57

^1^ Susceptibility to ampicillin, cefotaxime, ceftazidime, cefepime, cefmetazole, imipenem, meropenem, gentamicin, amikacin, levofloxacin, or ciprofloxacin.

**Table 3 microorganisms-11-01250-t003:** The difference in the signal intensity with or without CLA.

MIC (μg/mL)	Number of Strains	*I*_370_(CTX) − *I*_370_(CTX/CLA)
2	1	0.53
4	2	0.43
8	7	0.42 ± 0.12
32	2	0.50
64	25 ^1^	0.56 ± 0.19
	27	0.70 ± 0.38

^1^ Two strains showing a decrement value above 1.6 were excluded.

## Data Availability

Not applicable.

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
