# Peer review of "MALDI-TOF MS Approaches for the Identification of the Susceptibility of Extended-Spectrum β-Lactamases in Escherichia coli"

_microorganisms, 2023, doi:10.3390/microorganisms11051250_

Round 1
Reviewer 1 Report
The paper discusses about the detection of ESBL producers using MALDI-TOF MS within 15 minutes. I think that this is the most important moment of this whole study, because time is the essence in clinical microbiology. However, I have several important comments about the work and they are as follows:
1. The introduction is written too comprehensively and there is information at the beginning that I would remove and replace with information about ESBLs in E. coli. I see no reason for describing general information regarding resistance in S. aureus and the like. Focus the introduction specifically on your issue. In the same way, I would not mention your results right away in the beginning, but with gradual steps and subsequent information I got to your main goal, which was to detect the mechanism of resistance in E. coli to CTX and ESBL production using MALDI-TOF MS.
2. state exactly which ATCC and NCTC strain was resistant and which was susceptible – line 94
3. it is necessary to indicate the exact turbidity value for the repeatability of your results = line 97
4. what water? Line – 109
5. primers should be better to present in the table with citation – line 111-114
6. I can't find any information about DPD-1 plate with CTX and CTX/CLA, can you provide some sources for this test? – line 126
7. in similar experiments, in addition to hydrolysis, alkaline hydrolysis of the antibiotic is used as a comparative model, also with cefotaxime. why didn't you use it in your work?
8. the main parameters of the device setting are also not described, i mean for example: gain detector, laser atenuator, laser intenzity, laser frequency and other parameters.
9. the methodology of working with software equipment is also not described. there is no mention of collecting spectra, the number of laser shots, and no software in which the spectra were analyzed.
10. there is no mention of the analysis of pure compounds of the antibiotic CTX, which could have served as well as a calibrator together with the CHCA monomer and its dimer used in the work. The appearance of pure antibiotic without degradation products would be a clear signal that other peaks with the same molecular weight do not appear by chance in subsequent spectra. These could then cause a false positive interpretation of the results.
11. that is, in short, it would be appropriate to supplement the analysis of the ctx antibiotic using MALDI TOF, to supplement the analysis of alkaline hydrolysis as a comparative model, and to supplement the information about the software used in the work to evaluate and acquire the spectra.
12. Figure 1 – in description a molecular weight and exact compunds are missing. Also comparative pure antibiotic sepktra is missing.
13. Discussion must be improved and extended, is there a lot of information about MALDI TOF detection of degradation products of antibiotics. Also information about time consumption are available.
14. 12 citations is not enough for this type of study, please add more.
Author Response
Dear Reviewer, 1,
We are very grateful for your constructive comments and suggestions to our paper entitled “MALDI-TOF MS approaches for identification and susceptibility of the extended spectrum β-lactamases in Escherichia coli”. We have provided a point-by-point reply to the comments, and we have incorporated the related changes in the manuscript. We thank you for your thoughtful insights which helped to significantly improve the manuscript.
We have incorporated changes in the manuscript to reflect the suggestions provided by the reviewers. We have highlighted the changes within the manuscript.
- The introduction is written too comprehensively and there is information at the beginning that I would remove and replace with information about ESBLs in E. coli. I see no reason for describing general information regarding resistance in S. aureusand the like. Focus the introduction specifically on your issue. In the same way, I would not mention your results right away in the beginning, but with gradual steps and subsequent information I got to your main goal, which was to detect the mechanism of resistance in E. colito CTX and ESBL production using MALDI-TOF MS.
-> Thank you for your important suggestion. According to your suggestion, we replace the information at the bigining to the information about ESBLs in E. coli. Also we reviced the introduction part for our main goal. ( delete:line 33-35, 48-49, 65-71, 75-77. add: line 49-51, 54-58, 87-88)
- state exactly which ATCC and NCTC strain was resistant and which was susceptible – line 94
-> We are sincerely sorry; it was a mistake. We correct this point (line 114)
- it is necessary to indicate the exact turbidity value for the repeatability of your results = line 97
->Thank you for your suggestion. We added the turbidity value. (line 118)
- what water? Line – 109
-> We used the PCR grade water. We added in the manuscript. (line 139)
- primers should be better to present in the table with citation – line 111-114
->Thank you for your suggestion. We presented the information about primers in Table 1.
- I can't find any information about DPD-1 plate with CTX and CTX/CLA, can you provide some sources for this test? – line 126
-> Thank you foru your comment. DPD-1 plate is stopped selling on December 2022. So the information about DPD-1 is not available. We added it in the supporting information.
- in similar experiments, in addition to hydrolysis, alkaline hydrolysis of the antibiotic is used as a comparative model, also with cefotaxime. why didn't you use it in your work?
-> Thank you for your suggestion. It is useful for better understanding the hydrolysis of CTX to use alkaline hydrolysis of CTX as a cpmparative model. In 1994, Blanco F.R. et al. were reported a kinetic of the alkaline hydrolysis of cefotaxime using 1HNMR and HPLC and determined the process of hydrolysis mechanism (J. Pharm. Sci, 1994. 83(3), 322-327). In the field of clinical microbiology, there are many articles including review articles about the enzymatic activity using MALDI-TOF MS and the hydrolysis of CTX by beta-lactamase was well known. Moreover, we recently showd a reaction mechanism after hydrolysis of cefotaxime using MS/MS analysis by MALDI-TOF MS and flow injection analysis using LCMS-IT-TOF(Nikka Medical Association Magazine 2018. 67(1), 29-37.). Therefore, we did not use alkaline hydrolysis of CTX.
- the main parameters of the device setting are also not described, i mean for example: gain detector, laser atenuator, laser intenzity, laser frequency and other parameters.
-> Thank you for your suggestion. We added the information. (line 163-168)
- the methodology of working with software equipment is also not described. there is no mention of collecting spectra, the number of laser shots, and no software in which the spectra were analyzed.
-> Thank you for your suggestion. We added the information. (line 163-168)
- there is no mention of the analysis of pure compounds of the antibiotic CTX, which could have served as well as a calibrator together with the CHCA monomer and its dimer used in the work. The appearance of pure antibiotic without degradation products would be a clear signal that other peaks with the same molecular weight do not appear by chance in subsequent spectra. These could then cause a false positive interpretation of the results. that is, in short, it would be appropriate to supplement the analysis of the ctx antibiotic using MALDI TOF, to supplement the analysis of alkaline hydrolysis as a comparative model, and to supplement the information about the software used in the work to evaluate and acquire the spectra.
-> Thank you for your suggestion. As mentiond in the coment No.7, we think there is no need for using alkaline hydrolysis. Previousely we checked the Mass specturm by AXIMA Condfidence (Shimadzu, Japan) as a MALDI-TOF MS for research analysis using the same sample for measuring by VITEK MS Plus. The mass accuracy of VITEK MS PLUS used in this study seems not so good. This is because the accurate mass is slightly missing.
Exact Mass: 396.5 Da, Accurate mass: 396.01Da to 397.64 Da; Exact Mass: 370.5 Da, Accurate mass: 367.87 Da to 370.11 Da; Exact Mass: 414.5 Da, Accurate mass: 412.36 Da to 415.44 Da; Exact Mass: 456.5 Da, Accurate mass: 455.62 to 456.40 Da; Exact Mass: 478.5 Da, Accurate mass: 477.65 Da to 478.43 Da.
- Figure 1 – in description a molecular weight and exact compunds are missing. Also comparative pure antibiotic sepktra is missing.
-> Thank you for your suggestion. We added the exact mass in the main text.(line 287-297). As mentiond above, the accurate mass is slightly missing by the error contained in the VITEK MS Plus. We used the spectra of CTX in saline as the reference for unhydrolysis CTX .
- Discussion must be improved and extended, is there a lot of information about MALDI TOF detection of degradation products of antibiotics. Also information about time consumption are available.
-> Thank you for your suggestion. We added the information about the degradation products of CTX and time consumptions. (line 287-297, 303-311)
- 12 citations is not enough for this type of study, please add more.
-> Thank you for your suggestion. We added 12 references.
Other changes
->We apologized to the low quality of English. We used the Language Editing Services of MDPI before resubmitted the manuscript.
We believe that this study provides important new experimental data therefore for us the paper can be accepted in its current form.
We thank the reviewer for their comment and for their support regarding the publication of our work.

Reviewer 2 Report
Overall, your paper on the rapid detection of ESBL-producing bacteria using MALDI-TOF MS is well-conducted and presents valuable findings. However, I have a few suggestions for revision:
· Provide more detail: In some areas, the paper lacks detail that would help readers better understand the study. For example, it would be helpful to know more about the sample size and selection criteria for the clinical isolates used in the study.
· Include limitations: It would be useful to include a section on the limitations of the study. This would provide readers with a better understanding of the scope and generalizability of your findings.
· Consider revising the abstract: The abstract is a crucial part of the paper, as it is often the only part that readers will read. However, it is currently difficult to follow and does not provide a clear overview of the study.
· Improve the figures: Some of the figures in the manuscript are difficult to read and could benefit from better labeling and higher resolution.
· The bacterial names throughout the manuscript should be revised and write in italic.
Overall, your paper presents valuable findings and contributes to the field of microbiology. With these revisions, I believe that your paper will be even stronger and more accessible to a wider audience.
Introduction:
Lines 30-33: clarify the term "bactericidal strains," since this is not a commonly used term and could be confusing.
Line 46: define the abbreviation "MIC" (minimum inhibitory concentration) when it is first introduced.
Line 53-54: In the introduction, it might be helpful to clarify what is meant by "de-escalation therapy."
Methodology:
How were the clinical isolates of E. coli classified as β-lactamase-producing strains and ESBLs phenotype strains?
Line 132: “Four ul” Please correctly. I think it should be 4ul not four ul
What was the turbidity of the suspension adjusted to for the disk diffusion method and PCR-based detection?
What was the melting temperature obtained from analysis of the melting curve using light cycler nano software?
Results:
How were the clinically isolated microorganisms obtained and identified?
Can you explain how the detection of CTX and hydrolyzed CTX by VITEKⓇ MS Plus using standard strains works in more detail?
Can you clarify what the ratios in line 163 represent and how they were calculated?
Discussion:
In line 254, "using" is misspelled as "uisng".
In line 291, you mentioned that three strains showed the highest enzymatic activity. Can you provide more information on these strains, such as their MIC values and CTX-M gene expression levels?
In line 312, you suggested that further experiments need to be carried out. What specific experiments do you plan to do next and why?
Clarify technical language: While the paper is written in technical language, there are some parts that are unclear or difficult to understand. Please consider revising these sections to ensure that readers can follow your methods and results.
· Proofread the manuscript: There are several grammatical errors and typos in the manuscript that could be corrected through careful proofreading.
Author Response
Dear Reviewer, 2,
We are very grateful for your constructive comments and suggestions to our paper entitled “MALDI-TOF MS approaches for identification and susceptibility of the extended spectrum β-lactamases in Escherichia coli”. We have provided a point-by-point reply to the comments, and we have incorporated the related changes in the manuscript. We thank you for your thoughtful insights which helped to significantly improve the manuscript. We have incorporated changes in the manuscript to reflect the suggestions provided by the reviewers. We have highlighted the changes within the manuscript.
- Provide more detail: In some areas, the paper lacks detail that would help readers better understand the study. For example, it would be helpful to know more about the sample size and selection criteria for the clinical isolates used in the study.
-> Thanks, for your suggestion. The sample size was listed in Table 1. We added the information about the selection criteria were added. (line 174-175)
- Include limitations: It would be useful to include a section on the limitations of the study. This would provide readers with a better understanding of the scope and generalizability of your findings.
->Thank you for your kind suggestion. We added the limitation at the end of the discussion section. (line 403-405).
- Consider revising the abstract: The abstract is a crucial part of the paper, as it is often the only part that readers will read. However, it is currently difficult to follow and does not provide a clear overview of the study.
->Thank you for your suggestion. We revised the abstract.
- Improve the figures: Some of the figures in the manuscript are difficult to read and could benefit from better labeling and higher resolution.
-> Thank you for your suggestion. We checked all figures and figure legends, and revised.
- The bacterial names throughout the manuscript should be revised and write in italic.
-> We apologized for missing the bacterial names. We checked and wrote in italic form.
Overall, your paper presents valuable findings and contributes to the field of microbiology. With these revisions, I believe that your paper will be even stronger and more accessible to a wider audience.
Introduction:
Lines 30-33: clarify the term "bactericidal strains," since this is not a commonly used term and could be confusing.
-> We apologized for miss spelling. “bactericidal strains” was changed to “bacterial strains”.
Line 46: define the abbreviation "MIC" (minimum inhibitory concentration) when it is first introduced.
->Thank you. We spell out at this point.
Line 53-54: In the introduction, it might be helpful to clarify what is meant by "de-escalation therapy."
->Thank you for your suggestion. We revised that point to clarify the meaning of “de-escalation”.(line 54-58)
Methodology:
How were the clinical isolates of E. coli classified as β-lactamase-producing strains and ESBLs phenotype strains?
-> We identify the clinical isolates by using MALDI-TOF MS, classified as beta-lactamase producing strains by PCR assay, and classified ESBLs phenotype by disc diffusion method. We reviced the methodology in the Materials and Method section.(line119, 124-125)
Line 132: “Four ul” Please correctly. I think it should be 4ul not four ul
->We revised. “Four” to “4”. However, the “4” is changed to “four” by MDPI English editing service.
What was the turbidity of the suspension adjusted to for the disk diffusion method and PCR-based detection?
-> For the disk diffusion method, the turbidity of suspension was adjusted to McFarland standard No. 0.5 using saline. (line 118). For the PCR-based detection, the turbidity of suspension was adjusted to McFarland standard No. 3 using PCR grade water. (line 131)
What was the melting temperature obtained from analysis of the melting curve using light cycler nano software?
-> The melting temperature of the strains having blaCTX-M genes were 85-86℃for the ESBLs positive strains as reported by Nass T. et al. (line 145-147)
Results:
How were the clinically isolated microorganisms obtained and identified?
-> The clinically isolates were identified by MALDI-TOF MS. We added this information. (line 119)
Can you explain how the detection of CTX and hydrolyzed CTX by VITEKⓇ MS Plus using standard strains works in more detail?
->The information about preparing the analyzing samples were written in the “2.2 Hydrolysis assay” section. And the information about MALDI-TOF MS analysis was in the “2.3 MALDI-TOF MS analysis” section. We received the “2.3 MALDI-TOF MS analysis” section more detail. (line 164-168)
Can you clarify what the ratios in line 163 represent and how they were calculated?
->Thank you for your suggestion. We revised more detail. (line 203-205). We missed the signal at 396 Da. We recalculated the ratio and revised.
Discussion:
In line 254, "using" is misspelled as "uisng".
->We revised. “uisng” to “using”.
In line 291, you mentioned that three strains showed the highest enzymatic activity. Can you provide more information on these strains, such as their MIC values and CTX-M gene expression levels?
->In Figure 5, all strains showed their MIC value above 64 μg/mL and having the same CTX-M gene. As the gene expression does not equally mean the producing enzyme concentration, we did not check the expression levels.
In line 312, you suggested that further experiments need to be carried out. What specific experiments do you plan to do next and why?
->Thank you for your suggestion. We added at the end of the discussion section (line 403-410).
Comments on the Quality of English Language
Clarify technical language: While the paper is written in technical language, there are some parts that are unclear or difficult to understand. Please consider revising these sections to ensure that readers can follow your methods and results.
- Proofread the manuscript: There are several grammatical errors and typos in the manuscript that could be corrected through careful proofreading.
We believe that this study provides important new experimental data therefore for us the paper can be accepted in its current form.
->We apologized to the low quality of English. We used the Language Editing Services of MDPI before resubmitted the manuscript.
We thank the reviewer for their comment and for their support regarding the publication of our work.

Round 2
Reviewer 1 Report
Dear colleague,
i mean that all my suggestion was incorporated into the text. Thank you. Article can be publish.
Reviewer 2 Report
As a reviewer, I would like to congratulate the authors for revising the manuscript "MALDI-TOF MS approaches for identification and susceptibility of the extended spectrum beta-lactamases in Escherichia coli" in an exemplary manner. The changes made to the manuscript have significantly improved the clarity and organization of the content, making it easier to understand and follow.
some minor check will be good for spellings.